# Functional Whole Genome Screen of Nutrient-Starved *Mycobacterium tuberculosis* Identifies Genes Involved in Rifampin Tolerance

**DOI:** 10.3390/microorganisms11092269

**Published:** 2023-09-09

**Authors:** William M. Matern, Harley T. Harris, Carina Danchik, Marissa McDonald, Gopi Patel, Aashish Srivastava, Thomas R. Ioerger, Joel S. Bader, Petros C. Karakousis

**Affiliations:** 1Department of Biomedical Engineering, Institute for Computational Medicine, School of Medicine, Johns Hopkins University, Baltimore, MD 21205, USA; maternwill@gmail.com (W.M.M.);; 2Center for Systems Approaches to Infectious Diseases (C-SAID), School of Medicine, Johns Hopkins University, Baltimore, MD 21205, USA; hparke13@jhmi.edu (H.T.H.);; 3Tuberculosis Research Advancement Center, Department of Medicine, School of Medicine, Johns Hopkins University, Baltimore, MD 21205, USA; 4Department of Biochemistry and Biophysics, Texas A&M University, College Station, TX 77843, USA; srivastava@tamu.edu; 5Department of Computer Science and Engineering, Texas A&M University, College Station, TX 77843, USA; 6Department of International Health, Johns Hopkins Bloomberg School of Public Health, Baltimore, MD 21205, USA; 7Department of Molecular Microbiology and Immunology, Johns Hopkins Bloomberg School of Public Health, Baltimore, MD 21205, USA

**Keywords:** *Mycobacterium tuberculosis*, tolerance, persistence, virulence, transposon sequencing, molecular genetics, bioinformatics

## Abstract

*Mycobacterium tuberculosis* (*Mtb*), the causative agent of tuberculosis (TB), poses a global health challenge and is responsible for over a million deaths each year. Current treatment is lengthy and complex, and new, abbreviated regimens are urgently needed. *Mtb* adapts to nutrient starvation, a condition experienced during host infection, by shifting its metabolism and becoming tolerant to the killing activity of bactericidal antibiotics. An improved understanding of the mechanisms mediating antibiotic tolerance in *Mtb* can serve as the basis for developing more effective therapies. We performed a forward genetic screen to identify candidate *Mtb* genes involved in tolerance to the two key first-line antibiotics, rifampin and isoniazid, under nutrient-rich and nutrient-starved conditions. In nutrient-rich conditions, we found 220 mutants with differential antibiotic susceptibility (218 in the rifampin screen and 2 in the isoniazid screen). Following *Mtb* adaptation to nutrient starvation, 82 mutants showed differential antibiotic susceptibility (80 in the rifampin screen and 2 in the isoniazid screen). Using targeted mutagenesis, we validated the rifampin-hypersusceptible phenotype under nutrient starvation in *Mtb* mutants lacking the following genes: *ercc3*, *moeA1*, *rv0049*, and *rv2179c*. These findings shed light on potential therapeutic targets, which could help shorten the duration and complexity of antitubercular regimens.

## 1. Introduction

*Mycobacterium tuberculosis* (*Mtb*) is an acid-fast bacillus responsible for over 10 million cases of tuberculosis (TB) and approximately 1.5 million deaths annually, making it one of the leading causes of death by a single infectious agent worldwide [1]. The current standard treatment regimen for drug-susceptible TB requires 6–9 months of antibiotics: 2 months of daily rifampin, isoniazid, pyrazinamide, and ethambutol, followed by 4–7 months of rifampin and isoniazid. Because of the length and complexity of this treatment, medical nonadherence is a serious problem contributing to the emergence of antibiotic-resistant strains, for which treatment options are limited. Despite recent promising advances in the treatment of drug-resistant TB, including new antimicrobial regimens and bacteriophage therapy [2,3,4,5], progress in identifying new drug targets for TB treatment has been slow. Thus, only three new drugs from two different drug classes have been approved by the FDA in the past 40 years [6].

The duration and complexity of current antibiotic treatment regimens may be explained by the presence of antibiotic-tolerant bacterial subpopulations. Antibiotic tolerance is a phenotypic state in which bacteria become less susceptible to antibiotic killing due to reduced or arrested growth and/or altered metabolism. This manifests as increased time to bacterial killing for a given concentration of antibiotic [7]. Unlike drug resistance, drug tolerance is a transient and nonheritable phenomenon, which may arise from various virulence mechanisms, including reduced bacterial replication or growth arrest, the induction of efflux pumps, or altered metabolism [8,9,10]. Multiple stress conditions induce *Mtb* drug tolerance, including exposure to antibiotics, hypoxia, low pH, nutrient starvation, phosphate limitation, and macrophage defense mechanisms [11,12,13,14]. Directly targeting *Mtb* antibiotic tolerance mechanisms could promote more rapid bacterial clearance and decrease the time required for achieving a stable cure, with important implications for healthcare costs, medical adherence, and drug resistance. Identifying *Mtb* mutants with antibiotic hypersusceptibility phenotypes may provide insight into the molecular mechanisms underlying antibiotic tolerance. Furthermore, the products of such “antibiotic tolerance genes” may represent promising new drug targets for shortening TB therapy.

In the current study, we have utilized a modified transposon sequencing (Tn-Seq) approach [15,16] to identify mutants defective in their ability to survive exposure to a range of concentrations of the two first-line antitubercular drugs, rifampin and isoniazid, during growth in nutrient-rich broth and under nutrient starvation. Nutrient starvation is a physiologically relevant stress condition encountered by *Mtb* within fibrotic tuberculous lesions in vivo [17] and it has been shown to induce *Mtb* antibiotic tolerance to isoniazid and rifampin in vitro [12,18] through the upregulation of stress response pathways, including the stringent response [17,19,20]. The findings of our genetic screen may provide insight into potential therapeutic targets for enhancing the susceptibility of *Mtb* to the key sterilizing drug rifampin, with the ultimate goal of developing adjunctive therapies to shorten the duration of curative treatment for drug-susceptible TB.

## 2. Materials and Methods

### 2.1. Strains

*Mtb* H37Rv-JHU [21] was used for generating the transposon mutant libraries and for targeted mutagenesis studies.

### 2.2. Generation of Deletions

Targeted deletions were generated using the ORBIT recombineering system, as previously described [22]. Briefly, pKM444 was electroporated into H37Rv and plated on 7H10 agar plates containing kanamycin. Individual colonies were isolated and expanded under selective pressure. Expression of the genes encoded on pKM444 was induced using anhydrotetracycline and cultures were made electrocompetent. A second round of electroporation was used to insert pKM464 and a target-specific oligo (see Appendix A). Deletions were confirmed by PCR and Sanger sequencing of the 5′ and 3′ ends of the plasmid insertion (Genetic Resources Core Facility, Johns Hopkins University, Baltimore, MD, USA). Deletion strains were cultured in 7H9 media with hygromycin to maintain selective pressure.

### 2.3. Media and Buffers

Wild-type *Mtb* was grown in Middlebrook 7H9 broth containing 0.05% Tween-80, 0.2% glycerol, and 10% OADC. Transposon studies in nutrient-rich conditions were conducted in supplemented 7H9 broth, as described above, without the addition of glycerol (7H9/G-). For nutrient-starvation conditions, phosphate-buffered saline with Tween-80 (PBST) was prepared by adding 1.25 mL of 20% Tween-80 to 500 mL PBS to a final concentration of 0.05% Tween80 and filter-sterilized. Mycobacteriophage (MP) buffer was made by combining autoclave sterilized components to final concentrations of 50 mM Tris-HCl (pH 7.5), 150 mM NaCl, 10 mM MgSO_4_, and 2 mM CaCl_2_. CTAB solution for DNA extraction was made by combining 100 mL water, 5.84 g NaCl, 1 g hexadecyltrimethylammonium bromide, and 0.2 mL 0.5 M EDTA. Antibiotics for selective pressure were used at the following concentrations unless indicated otherwise: kanamycin (20 μg/mL) and hygromycin (50 μg/mL). 7H11 agar was made using 10.25 g of Middlebrook 7H11 without malachite green powder (HiMedia, Kennett Square, PA, USA) added to 450 mL deionized water. 5 mL 50% glycerol was then added before autoclaving. The agar was cooled to 55 °C before addition of 50 mL OADC enrichment (Becton Dickinson, Franklin Lakes, NJ, USA).

### 2.4. Generation of Transposon Mutant Library

Transposon (Tn) mutant pools were constructed in H37Rv by adapting a previously published protocol [23] using ϕmycomarT7. *Mtb* strain H37Rv was grown from frozen stocks in 450 mL 7H9 media to OD_600_ 1.2 in a large roller bottle at 37 °C with shaking. Culture was split into 7 tubes of 50 mL/tube. Tubes were centrifuged (2000× *g* for 5 min) and resuspended in 10 mL MP buffer. This washing step was repeated two additional times. Cells were then centrifuged (2000× *g* for 5 min) once more and immersed in 4 mL of MP buffer. Approximately 1 × 10^11^ plaque-forming units (10:1 phage: bacilli) of ϕmycomarT7 were added to 6 of the tubes. The seventh tube received only MP buffer without phage as a control. Tubes were placed on a shaking incubator (37 °C) for two days. After incubation, the transformation mixture was centrifuged (2000× *g* for 5 min) and washed with PBST to remove residual phage. An additional centrifugation was performed, and bacteria were immersed in 1 mL of PBST.

50 μL of each tube of washed transformants (or no-vector control) were diluted and plated on 7H11 plates, with or without 50 μg/mL kanamycin, to determine transformation efficiency and background resistance. The remainder of the cultures were plated on 7H11 containing 50 μg/mL kanamycin in Pyrex baking dishes (15′′ × 10′′, 500 mL agar per dish, 1 tube per dish). After 35 days, colonies were scraped from each dish and dispersed by vortexing with sterile glass beads (3 mm) in fresh 7H9 broth. Equal volumes of all six pools were combined to increase mutant diversity and frozen in aliquots at −80 °C.

### 2.5. Setup for Tn Screen in Nutrient-Rich Broth

The overall schematic for the Tn screen is shown in Figure 1. A 1-mL aliquot of the combined Tn mutant pool (consisting of ~1.2 × 10^6^ unique mutants) was inoculated into 200 mL of 7H9/G- in a 1.3-L roller bottle and shaken at 37 °C for 3 days to reduce bacterial clumping. These samples were then split into four 50-mL aliquots and centrifuged at 2000× *g* for 5 min. The resulting pellets were resuspended in 5 mL of 7H9/G- and pooled into a single tube. The culture was then strained through a 40-μm cell strainer to remove macroscopic clumps. The OD_600_ was measured, and the culture was diluted to OD 0.1 and aliquoted into 21 tubes (10 mL/tube) for the screen. After 1 day of shaking at 37 °C, drugs (dissolved in water) were added to each tube (day 0). Ranges of drug concentrations were selected based on estimates of the maximum concentration observed in human plasma, with subsequent dilutions to simulate reduced drug exposures. C_max_ was assumed to be 1.0 µg/mL for isoniazid and 4.0 µg/mL for rifampin based on available data [24], and ten-fold serial dilutions were performed. Final concentrations of isoniazid (INH) were 0.01 µg/mL, 0.1 µg/mL, and 1.0 µg/mL. Final concentrations of rifampin were 0.04 µg/mL, 0.4 µg/mL, and 4.0 µg/mL. Three replicates for each drug concentration were performed, as well as a no-drug control (21 tubes total). Cultures were checked daily for bacterial growth. If growth exceeded OD_600_ of 0.8, the culture was diluted 1:20 into fresh broth to keep the cultures in log-phase throughout the screen. Each day 400 μL of culture were removed and processed for colony-forming unit (CFU) enumeration. Briefly, samples were centrifuged (3000 g for 5 min) and washed twice with 7H9/G- to remove residual antibiotics. Ten-fold serial dilutions were performed, and 100 μL of each dilution was plated on 7H11 agar. CFU were counted after 34–36 days.

After 6 days of drug exposure, tubes were centrifuged (2000× *g* for 5 min) thrice and washed with 10 mL of 7H9/G- to remove residual antibiotics and cell debris before resuspending in a final volume of 10.8 mL for regrowth. An amount of 0.8 mL of each culture was then removed to confirm OD < 0.4 and diluted if appropriate. Tubes were passaged and diluted (1:20) in fresh 7H9/G- upon reaching OD ≥ 0.8. This was repeated twice (approximately 9–10 doublings). Regrown mutant pools were then centrifuged and processed for DNA extraction. 

Only a subset of samples was chosen to proceed with Tn-Seq and analysis. We considered the following criteria for antibiotic exposures to select groups for analysis: (1) Bacterial numbers must exceed 10^6^ CFU/mL at all times during exposure, ensuring that the probability of missing a particular mutant is minimized (Appendix A); (2) There should be detectable attenuation of bacterial viability after addition of the drug, ensuring the concentration of antibiotic is high enough to kill highly susceptible mutants. (Appendix A); (3) A preference for drug concentrations near or below achievable serum concentrations of drug after standard dosing.

### 2.6. Setup for Tn Screen in Nutrient Starvation

For screening the *Mtb* Tn mutant library in starvation conditions, we again inoculated a 1-mL aliquot of the combined Tn mutant pool into a 1.3-L roller bottle containing 200 mL of 7H9/G-. This was shaken at 37 °C for 1 day to reduce bacterial clumping. An amount of 50 mL of this culture was then split into 5–10-mL aliquots and incubated on a shaker for 1 day. The cultures were then pooled, diluted to an OD_600_ of 0.01, and aliquoted into 60 tubes (10 mL/tube) to reduce bacterial clumping. After cultures reached an OD_600_ of 0.85, they were combined, centrifuged (3000× *g* for 5 min), and washed twice with PBST. The culture was then diluted to OD_600_ of 0.52 in PBST and aliquoted into 90 tubes (10 mL/tube) to perform the screen. Immediately, 3 tubes were removed for CFU enumeration (day −17). After 17 days in PBST, an additional 3 tubes were removed for CFU enumeration and regrowth (day 0). Drugs (dissolved in water) were then added to the remaining tubes at the same concentrations as for the Tn screen in nutrient-rich broth. Six replicates were performed for each drug concentration and time point, as well as for no-drug controls (84 tubes total). Samples were taken for regrowth and CFU enumeration at days 7 and 14 of drug exposure.

CFU enumeration was performed as above with the addition of washing once with PBST before dilution and plating on 7H11 agar. CFU were counted after 25–35 days. For regrowth, the remainder of each tube was centrifuged (3000× *g* for 5–10 min) and washed with 10 mL of PBST twice to remove drugs. After final centrifugation, the sample was resuspended in 250 μL of PBST. An amount of 50 μL of each washed Tn mutant pool was plated on four 7H11 agar plates and grown until a bacterial lawn was formed (7–14 days, depending on group). Bacterial lawns were scraped and pooled into 2-mL tubes for DNA extraction.

### 2.7. DNA Extraction

For DNA extraction, a pellet of *Mtb* was obtained in an O-ring tube and the supernatant was removed. The pellet was then heated to 85 °C for 20–30 min. After heat-killing, the pellet was resuspended in 0.6 mL of CTAB extraction solution. The sample was then transferred into a 2 mL tube containing 1 g of 0.1 mm zirconia beads and 0.6 mL of chloroform. The sample was then bead-beaten for 30 s at 7200 rpm on a Presellys Evolution (Bertin Technologies, Montigny-le-Bretonneux, France) before centrifugation at 16,000× *g* for 2 min. Approximately 0.4 mL of the aqueous (top) phase was then pipetted into a fresh tube, avoiding the white interface. Two volumes of 100% ethanol were added, and the sample was mixed by inversion to precipitate DNA. In cases where DNA was visible by eye, a 2 min centrifugation was performed, otherwise centrifugation was extended to 15 min. Supernatant was removed and DNA pellet was washed with 70% ethanol (700 μL). The sample was briefly centrifuged again, and the ethanol was removed. The pellet was allowed to dry for 15 min before resuspension in Tris-Cl (100 μL). A NanoDrop (ThermoFisher, Waltham, MA, USA) was used to quantify DNA concentration.

### 2.8. DNA Library Preparation

DNA library prep was performed as previously described [25]. DNA libraries were sequenced (2 × 100 bp) on an Illumina HiSeq 2500 (Illumina Inc., San Diego, CA, USA) by Tom Ioerger and Aashish Srivastava of Texas A&M University.

### 2.9. Hypothesis Testing for Antibiotic Hypersusceptibility

Calculations of log_2_-fold-change (LFC) and adjusted *p*-values for each condition were performed using a previously described approach [26]. This method is especially applicable to experiments containing more than one concentration of a drug, as was performed here with rifampin in nutrient-rich broth and under nutrient starvation. In such cases, we expect that mutants hypersusceptible or hypertolerant to a drug will be even more hyper-susceptible/tolerant at higher doses, which enables a statistical test that increases statistical power (Jonckheere-Terpstra test). For the screen in nutrient-rich broth, Tn mutants were considered differentially susceptible to a drug if the absolute value of the LFC relative to the no-drug control was >0.5 and the adjusted *p*-value (Benjamini–Hochberg) was <0.05. For the starvation medium screen, Tn mutants were considered differentially susceptible to a drug if the absolute value of the LFC relative to the no-drug control was >0.5 and the adjusted *p*-value (Benjamini–Hochberg) was <0.05 at both the 7 and 14 day time points. For further comparison and analysis of this dataset, the 0.04 μg/mL rifampin condition was used for nutrient-rich broth and 4.0 μg/mL rifampin was used for data derived from the nutrient-starvation model. Data from isoniazid 1.0 μg/mL were used for analysis in both conditions. 

### 2.10. Oligonucleotides and Primers

Targeting oligonucleotides were designed as described by Murphy et al. [22]. Briefly, constructs contained 37 base pairs upstream and downstream of the gene of interest and the first and last 30 coding nucleotides of the lagging strand of each gene connected by the attP sequence GGTTTGTCTGGTCAACCACCgcggtctcAGTGGTGTACGGTACAAACC. All oligonucleotides and primers were ordered from IDT with standard desalting and resuspended to 100 μM in TE buffer. Oligos and primers are listed in Appendix A.

### 2.11. Plasmids 

The plasmids required for targeted gene deletion utilizing the ORBIT recombineering system [22] are described here. pKM444 was a gift from Kenan Murphy (Addgene plasmid # 108319; http://n2t.net/addgene:108319 (accessed on 8 September 2023); RRID: Addgene_108319). pKM444 expresses Che9c phage RecT and Bxb1 phage integrase proteins under tetracycline induction. This plasmid also contains a kanamycin resistance cassette for selection. pKM464 was a gift from Kenan Murphy (Addgene plasmid # 108322; http://n2t.net/addgene:108322 (accessed on 8 September 2023); RRID: Addgene_108322). pKM464 is the ORBIT integrating plasmid for gene deletion and contains a hygromycin resistance cassette for selection. 

### 2.12. Time-Kill Assays in Nutrient-Rich Broth

Mid-log phase cultures were diluted to an OD_600_ of 0.1 and split into 6 tubes per culture. Rifampin was added to half the tubes at a final concentration of 2 μg/mL and an equal volume of DMSO vehicle was added to the other half. Aliquots were taken for plating on 7H11 plates at days 0, 2, 4, and 6. Inoculated plates were incubated for 3 weeks before CFU enumeration. 

### 2.13. Time-Kill Assays in Nutrient Starvation Conditions

Cultures were grown to late log phase (OD_600_~0.8), then centrifuged and resuspended in an equal volume of PBST twice before being diluted to an OD_600_ of 0.5. Tubes were incubated without shaking for 2 weeks before separation into 6 tubes per culture and drug exposure. Rifampin (2 μg/mL) or DMSO vehicle was added to each tube and aliquots were taken for plating on 7H11 agar at days 0, 3, and 6. Agar plates were incubated for 3 weeks before CFU enumeration.

## 3. Results

### 3.1. Overview of Isoniazid and Rifampin Screens of Mtb Transposon Mutant Libraries

A total of six separate saturated Tn mutant libraries were generated in the lab reference strain *Mtb* H37Rv. These libraries were screened for genes required for tolerance to a range of concentrations of isoniazid or rifampin, selected based on human pharmacokinetic data [24], in nutrient-rich broth (7H9) or under nutrient starvation (phosphate-buffered saline (PBS)) (Figure 1). In nutrient-rich broth, mutants containing Tn insertions in *cinA*/*rv1901* and *ahpC*/*rv2428* (*rv1901*::Tn and *rv2428*::Tn, respectively) were identified as being hypersusceptible to isoniazid, whereas none were found to have reduced susceptibility to the drug. Insertion mutants in 218 genes were identified as being differentially susceptible to rifampin (126 hypersusceptible and 92 hypertolerant). During nutrient starvation, two mutants were identified as being hypersusceptible to isoniazid (*rv1901*::Tn and *rv0767c*::Tn), and a total of 80 mutants were identified with differential susceptibility to rifampin (41 hypersusceptible and 39 hypertolerant). A summary of these results is plotted in Figure 2. A complete list of mutants with statistically significant phenotypes and their effect sizes by drug, dose, and time point can be found in Appendix A.

### 3.2. Identification of Mtb Genes Contributing to Rifampin Tolerance in Nutrient-Rich Broth and under Nutrient Starvation

To identify conserved *Mtb* genes with a potential role in rifamycin tolerance across mycobacteria, we compared the results of the *Mtb* mutant library screen exposed to rifampin in nutrient-rich broth with our previously published *M. avium* mutant screen exposed to sublethal concentrations of rifabutin in nutrient-rich broth [26]. We identified eight homologous gene pairs whose disruption was associated with significant rifamycin hypersusceptibility and 1 gene pair whose disruption resulted in significant rifamycin hypertolerance (|log_2_-fold change| (|LFC|) > 1) in both *Mtb* and *M. avium* (Table 1). Negative and positive LFC values indicate rifamycin hypersusceptibility and hypertolerance, respectively. 

To explore the role environmental stress plays in *Mtb* antibiotic tolerance, we compared the data from our Tn mutant screens in nutrient-rich vs. nutrient-starved conditions. We identified two categories of mutants with significant phenotypes: those which responded similarly under both screening conditions and those which responded differently. Table 2 lists Tn mutants with differential susceptibility to rifampin only under nutrient starvation, but with minimal or opposite differential susceptibility in nutrient-rich broth. Mutants included in the list are those found to have significantly (|LFC| > 1) altered susceptibility to rifampin following 14 days of nutrient starvation, but unchanged rifampin susceptibility (|LFC| < 0.5) in nutrient-rich broth. For example, *rv0458*::Tn was found to be hypertolerant to rifampin during nutrient starvation, but its rifampin susceptibility was similar to wild-type (near-zero LFC) in nutrient-rich broth.

Table 3 lists mutants with differential susceptibility to selected concentrations of rifampin under both nutrient starvation and nutrient-rich conditions (4.0 μg/mL rifampin and 0.04 μg/mL rifampin, respectively). To compute this list, mutants with a statistically significant phenotype and effect size > 1 in either direction (|LFC| > 1) during nutrient starvation were first identified. These mutants were then examined in data collected from nutrient-rich broth. Those genes with |LFC| > 1 in the same direction as the data from the nutrient-starvation screen (i.e., hypertolerant or hypersusceptible) were included.

We decided to further investigate those *Mtb* genes whose disruption was associated with significantly increased (LFC < −1; adjusted *p*-value < 0.05) susceptibility to rifampin in both conditions (Table 3). Three genes met these criteria, namely *rv0049*, *moeA1*/*rv0994*, and *rv2179c*. *rv0049*::Tn had a rifampin LFC of −2.99 (*p =* 1.40 × 10^−8^) in nutrient-rich broth and LFC values of −1.45 (*p =* 6.12 × 10^−15^) and −1.81 (*p =* 2.03 × 10^−19^) after rifampin exposure for 7 and 14 days, respectively, under nutrient starvation. *moeA1*/*rv0994*::Tn had a rifampin LFC of −3.25 (*p* = 1.60 × 10^−5^) in nutrient-rich broth and LFC values of −1.27 (*p* = 2.18 × 10^−7^) and −1.01 (*p =* 1.31 × 10^−5^) after rifampin exposure for 7 and 14 days, respectively, under nutrient starvation. *rv2179c*::Tn had a rifampin LFC of −2.72 (*p* = 1.39 × 10^−7^) in nutrient-rich broth and LFC values of −1.32 (*p* = 6.24 × 10^−18^) and −1.3 (*p* = 3.22 × 10^−19^) after rifampin exposure for 7 and 14 days, respectively, under nutrient starvation (Table 2). We also included *ercc3* (*rv0861c*) as a gene of interest, since *ercc3*::Tn had a LFC of −1.45 (*p* = 8.58 × 10^−13^) in nutrient-rich broth and smaller, but still significant, LFC values of −0.76 (*p* = 1.92 × 10^−9^) and −0.63 (*p* = 4.46 × 10^−10^) after 7 and 14 days, respectively, of rifampin exposure under nutrient starvation.

### 3.3. Validation of Specific Mtb Genetic Requirements for Rifampin Hypersusceptibility Using Targeted Mutagenesis

Gene-deficient recombinant strains for each of the four genes discussed above were generated using the ORBIT recombineering system [22] and confirmed via PCR and Sanger sequencing. To validate the findings of the nutrient-starvation screen, these gene-deficient strains were tested for rifampin susceptibility in time–kill assays under the same stress conditions. After 6 days of exposure, rifampin showed significantly more killing against nutrient-starved cultures of each of the four mutants tested compared to nutrient-starved wild-type *Mtb* (Figure 3A–E). Relative to the nutrient-starved wild-type strain, rifampin 2 μg/mL reduced the bacterial density of nutrient-starved cultures of Δ*ercc3*, Δ*moeA1*, Δ*rv0049*, and Δ*rv2179c* by 2.6-fold, 2.2-fold, 3.9-fold, and 4.0-fold, respectively (Figure 3E).

## 4. Discussion

For the first time, we have identified *Mtb* genes specifically responsible for reduced susceptibility to the first-line drug rifampin under nutrient starvation, a physiologically relevant condition known to induce antibiotic tolerance [17,19]. The four *Mtb* genes implicated in nutrient starvation-induced rifampin tolerance encode a DNA helicase (Ercc3), an enzyme involved in molybdenum cofactor synthesis (MoeA1), a hypothetical protein with an unknown function (Rv0049), and a homolog of *E. coli* RNase T (Rv2179c). Disruption of each of these four genes conferred hypersusceptibility to rifampin in both nutrient-rich and nutrient-starvation screens, suggesting that their products may be promising targets across different populations of *Mtb*, which may encounter a range of stress conditions and microenvironments within host tissues [11,12,13,14]. In addition to elucidating mechanisms of *Mtb* virulence, our findings pave the way for the development of novel anti-virulence agents aimed at enhancing the sterilizing activity of rifampin and accelerating the clearance of drug-susceptible TB.

Ercc3 is an ATP-dependent 3′ to 5′ DNA repair helicase homologous to the mammalian XPB; although it lacks the C-terminal domain [27]. Its eukaryotic counterpart plays roles in both transcription initiation and nucleotide excision repair; though the functions of the bacterial enzyme Ercc3 are less well-characterized [27]. It binds to both single-stranded and double-stranded DNA and has DNA-dependent ATPase activity; catalyzing DNA unwinding; as well as ATP-independent DNA-annealing capabilities [28]. No studies have previously linked this gene to antibiotic tolerance; though its likely roles in transcription and DNA repair suggest a link with rifampin’s known role as a transcription inhibitor. 

MoeA1/Rv0994 is involved in the biosynthesis of molybdenum cofactor (MoCo), a pathway highly conserved across organisms from different kingdoms of life [29]. While all mycobacteria can generate MoCo de novo, *Mtb* has an expanded number of genes in this pathway [30]. Multiple forward genetic screens in *Mtb* have identified the involvement of MoCo in ΤΒ pathogenesis [29]. Furthermore, reduced synthesis of MoCo via disruption of a gene cluster including *moeA1* resulted in reduced bacterial survival within macrophages [31]. MoeA1 may also be involved in antibiotic tolerance through its role as a target of DosS, the membrane-associated sensor histidine kinase of the two-component regulatory system DosR/S, which is required for persistence and maintenance of *Mtb* infection [32].

A crystal structure of Rv2179c previously revealed structural homology to *E. coli* RNase T despite low amino acid sequence similarity [33]. Like RNase T, Rv2179c has ATP-dependent 3′ exonuclease activity against single-stranded overhangs of dsRNA. It also has orthologs in all sequenced *Mycobacterium* species, suggesting that it is the first identified member of a novel family of RNases [33]. Rv2179c has been implicated in pathogenesis, as anti-Rv2179c IgG levels are higher in persons with active TB disease than in those with asymptomatic TB infection [34]. Although the potential role of Rv2179c in antibiotic tolerance remains to be characterized, there are other examples of RNases, such as the VapC toxins, which have been implicated in *Mtb* persistence through site-specific cleavage of tRNAs and translation inhibition [35,36].

Rv0049 is a hypothetical protein conserved across mycobacteria, including *Mtb*, *M. leprae*, *M. marinum*, *M. avium*, and *M. smegmatis* [37]. Rv0049 is upregulated under nutrient starvation in vitro [19] and is required for *Mtb* survival within murine bone marrow-derived macrophages [31], but its function remains unknown. While data are lacking on the potential role of Rv0049 in antibiotic tolerance, immune responses to this protein have been proposed as blood-based biomarkers. Thus, serum IgM antibodies to Rv0049 are elevated in patients with active TB, and their presence is able to accurately distinguish active TB from asymptomatic infection with a sensitivity of 52.1% and a specificity of 80.6% [34]. Based on the amino-acid sequence of Rv0049, a deep learning method for determining protein surfaces called dMaSIF (differentiable molecular surface interaction fingerprinting) [38,39,40] identified binding hot spots at residues 4, 7, 30–31, 42, 54, 56, 59, 61, 70, 91, 93–94, and 104–107.

We have previously performed a similar Tn-Seq screen in *M. avium complex* to identify genes required for antibiotic tolerance in that organism [26]. We compared the *Mtb* genes conferring differential susceptibility to rifampin to those previously shown to confer differential susceptibility to rifabutin in *M. avium* [26] (Table 1). Several of the putative rifampin tolerance genes encode proteins involved in phosphate transport, such as *rv0820* (*phoT*), *rv0929* (*pstA1*), and *rv0930* (*pstC2*). This suggests a connection between mycobacterial phosphate transport and rifamycin-mediated killing, an observation which has been reported previously [41,42] and attributed to constitutive activation of the phosphate starvation response [41], which is regulated by the two-component regulatory system, SenX3-RegX3 [14]. It is possible that low intrabacterial phosphate levels may boost rifamycin activity in mycobacteria. Of note, both the nutrient-rich and nutrient starvation (PBS) models used in this study are not phosphate-limiting. Alternatively, mycobacterial phosphate transporters may be involved in rifamycin efflux. The requirement of *rv2179c* for rifamycin tolerance of *Mtb* during nutrient starvation and of *M. avium* in nutrient-rich broth [26] may reflect its hypothetical role in processing functional RNA molecules. Additionally, the hypothetical proteins Rv0049, Rv1836c, and Rv3005c were also found to be involved in rifamycin tolerance in both *M. avium* and *Mtb*. The single gene disruption found to confer hypertolerance to rifamycins was *rv0819*, annotated as a mycothiol acetyltransferase. This is somewhat surprising given that mycothiol synthesis has been linked to rifamycin hypersusceptibility [43]. Notably, the gene *rv0819/DFS55_21365* is oriented such that a Tn disruption in this gene could affect the downstream transcription of the adjacent gene *rv0820/DFS55_21345*, one of the phosphate genes causing rifamycin hypersusceptibility, as discussed above. It has been noted previously that the kanamycin promoter within the *Himar1* Tn can serve as a promoter for downstream gene expression [44]. Therefore, it is possible that a Tn insertion in this gene promotes survival during exposure to rifamycins due to increased expression of Rv0820, which, as noted above, may contribute to mycobacterial survival upon rifamycin exposure.

Others have also utilized this methodology for investigating the intrinsic mechanisms of antibiotic susceptibility in *Mtb* under nutrient-rich [45] and phosphate-starvation [46] conditions. More recently, Li et al. used a CRISPR interference chemical-genetics platform to quantify *Mtb* fitness following gene depletion in the presence of different antibiotics, including rifampin [47]. Their screen identified 503 genes with decreased fitness to rifampin upon gene depletion, as compared to our study, which identified 126 genes involved in rifampin tolerance. The genes *moeA1 and Rv1836c* were identified as contributing significantly to intrinsic resistance to rifampin. In contrast, depletion of *ercc3, rv0049,* and *rv2179c* did not confer increased susceptibility to any concentration of rifampin studied. Discrepancies between our findings and those of Li et al. may be explained by the different methodologies employed in each study. In contrast to our transposon-based approach, which limits the investigation to genes that are not essential for *Mtb* growth in nutrient-rich conditions, the CRISPR interference method has the potential to identify additional antibiotic tolerance genes since it is able to target both essential and nonessential *Mtb* genes. On the other hand, the incomplete depletion of gene expression using the latter approach may help explain the fact that Li et al. did not identify *ercc3*, *rv0049*, and *rv2179c* in their rifampin screen, perhaps since very low levels of these gene products are sufficient to preserve the wild-type rifampin susceptibility phenotype. Our study revealed that these genes contribute to rifampin tolerance in both nutrient-rich and nutrient starvation conditions, while Li et al. used only nutrient-rich broth in their screen. Additionally, the lowest concentration of rifampin used by Li et al. was 0.66 μg/mL, which is more than ten-fold higher than the concentration used in the current study.

The recent screen by Block et al. evaluated antibiotic tolerance under two separate stress conditions (stationary phase and phosphate starvation) and screened ~1000 mutant subsets of a defined collection of Tn mutants in *Mtb* Erdman [46]. Their cutoffs were more stringent than ours (LFC > |2| and *p* < 0.025 versus LFC > |0.5| and *p* < 0.05). Despite the different *Mtb* strain, stress conditions, drug concentrations and combinations in each study, both our nutrient-starvation rifampin screen and the stationary-phase screen with rifampin/isoniazid by Block et al. identified a cluster of rifampin tolerance genes containing *mce1C*/*rv0171*, *mce1D*/*rv0172*, and *mce1F*/*rv0174*. One of the *mceD1* transposon mutants showed reduced survival in their ciprofloxacin + isoniazid stationary-phase conditions, and all others trended towards statistical significance in all drug and stress combinations tested. Mce proteins complex to form lipid or sterol transporters and have been implicated in *Mtb* pathogenesis through their influence on host signaling pathways [48].

Xu et al. have previously conducted a similar Tn screen with exposure to rifampin and isoniazid in nutrient-rich conditions [45]. Comparing our data from the nutrient-rich screen to those of Xu et al., we identified 42 genes in common that conferred differential susceptibility to rifampin. A total of 33 genes were identified by Xu et al. which did not appear as significant in our screen. Conversely, we found 81 unique *Mtb* mutants, which were differentially susceptible to rifampin that were not previously identified by Xu et al. It is possible that the discrepancy in these data is due to differences in experimental methodology and/or bioinformatics analyses. In each study, *Mtb* H37Rv was grown in supplemented Middlebrook 7H9 broth. However, the rifampin concentration utilized in each screen differed. Our screen used a concentration of 0.04 μg/mL, which is 10-fold higher than that used by Xu et al. Our analysis also utilized a different statistical method (Jonckheere-Terpstra test vs the TA-site permutation test implemented in TRANSIT), and we applied the additional log-fold change requirement |LFC| > 0.5 for genes to be considered significant. The overlap of shared genes is still highly significant, with a two-sided *p*-value from a 2 × 2 contingency table of 4.0 × 10^–32^ assuming 1000 genes tested, and a *p*-value of 6.2 × 10^–195^ assuming 5000 genes tested. Of note, our screen additionally looked at nutrient starvation, which is more biologically relevant due to similar stress conditions being encountered in the host environment [17].

Although we identified over one hundred *Mtb* Tn mutants with altered rifampin susceptibility relative to wild-type bacteria, only three mutants were detected as being hypersusceptible to isoniazid in nutrient starvation or nutrient-rich broth. While it is possible that the number of genes mediating tolerance to isoniazid is indeed lower than that for rifampin, it is more likely that in the nutrient-rich conditions the lowest concentration of isoniazid may have been insufficient, as evidenced by the lack of difference in bacterial density for bulk cultures of the group exposed to isoniazid and the untreated control (Appendix A). This may have reduced the effect size of mutants hypersusceptible to isoniazid to below our cutoffs for statistical significance. Additionally, a 10-fold and 100-fold higher concentration of isoniazid resulted in excessive *Mtb* killing, resulting in insufficient bacterial regrowth for the purposes of Tn-seq (Appendix A). Therefore, future studies may consider either using intermediate isoniazid concentrations or reducing the duration of isoniazid exposure before bacterial regrowth. This was performed successfully by Xu et al., who used an intermediate concentration of isoniazid and identified 54 differentially susceptible hits for a partial growth-inhibitory concentration of isoniazid in nutrient-rich broth [45].

Despite the difference in the number of total hits, both our screen and that of Xu et al. identified *cinA* (*rv1901*) as important for tolerance to isoniazid [45]. In concordance with both findings, subsequent studies found that a *ΔcinA* mutant had increased susceptibility to isoniazid during nutrient starvation, possibly related to the putative role of CinA in cleaving isoniazid-NAD adducts [49]. Deletion of *cinA* also led to increased *Mtb* killing within macrophages and increased activity of isoniazid against TB infection in mice, confirming the potential utility of targeting CinA as a novel therapeutic strategy [49].

In the current study, we have identified multiple *Mtb* genes which confer tolerance to rifampin under nutrient starvation. The products of these genes may serve as targets for future drug development campaigns, allowing for direct disruption of rifampin tolerance pathways and the shortening of TB treatment.

## Figures and Tables

**Figure 1 microorganisms-11-02269-f001:**
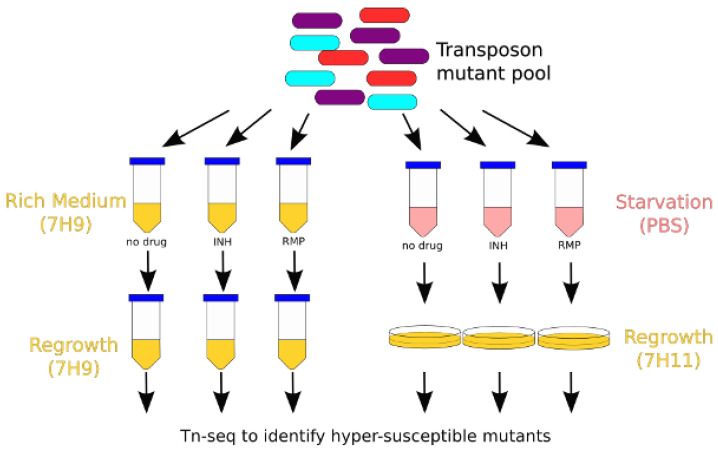
Schematic of experimental setup for identifying antibiotic tolerance genes. A transposon (Tn) mutant library generated in H37Rv was aliquoted into separate tubes. After a period of adjustment to the new conditions, isoniazid (INH) or rifampin (RMP) was added, followed by regrowth and DNA sequencing.

**Figure 2 microorganisms-11-02269-f002:**
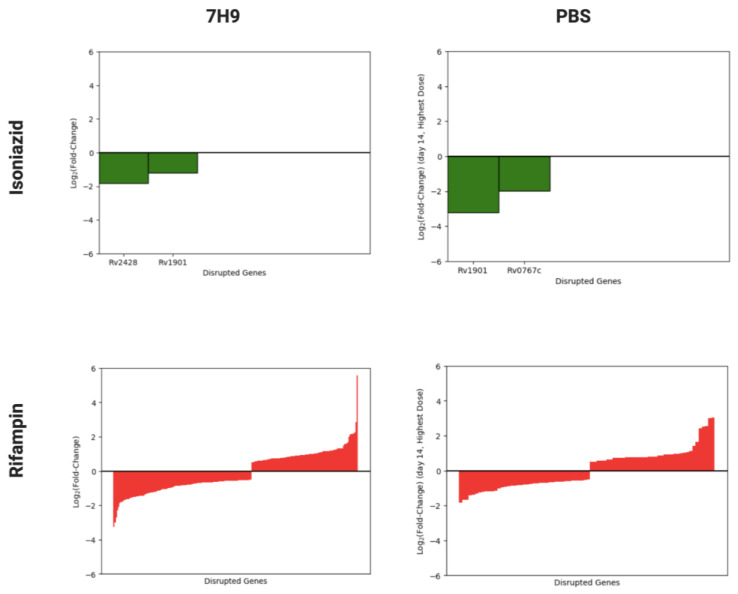
The effect of disruption of *Mtb* H37Rv genes on mycobacterial survival in nutrient-rich (7H9) and nutrient-starved (PBS) broth following exposure to sublethal concentrations of isoniazid (green) and rifampin (red). Log2 fold-change is normalized to untreated controls. Transposon disruption mutants are sorted from greatest to smallest effect size. Only statistically significant genes are plotted. Negative and positive values represent hypersusceptible and hypertolerant mutants, respectively.

**Figure 3 microorganisms-11-02269-f003:**
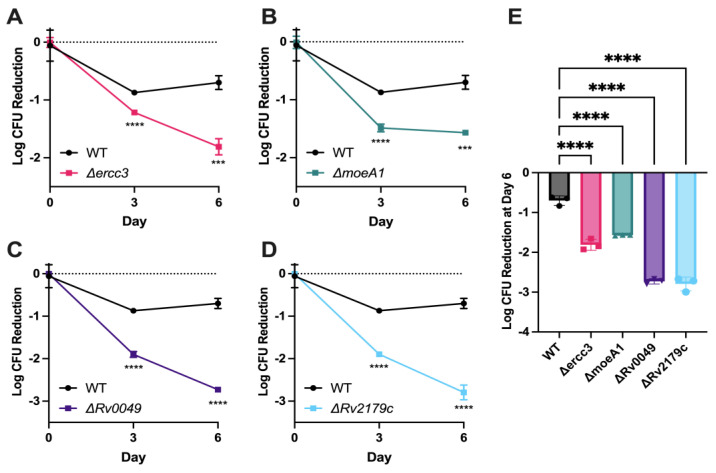
Time–kill assays of nutrient-starved *Mtb* recombinant strains following exposure to rifampin. Each culture was starved of nutrients for 2 weeks prior to exposure to rifampin at 4 μg/mL. (**A**) *Δercc3*; (**B**) *ΔmoeA1*; (**C**) *Δrv0049*; and (**D**) *Δrv2179c*. (**E**) Mean log_10_ colony-forming unit (CFU) reduction in each strain after 6 days of rifampin exposure, relative to their respective mean CFU counts on day 0. WT = Wild type. An ordinary one-way ANOVA test was followed by Dunnett’s multiple comparisons test to determine significance (*** = *p*-value < 0.001, **** = *p*-value < 0.0001). DMSO control samples time kill assays shown in Appendix A.

**Table 1 microorganisms-11-02269-t001:** *Mtb* and *M. avium (Mav)* transposon mutants with differential susceptibility to rifamycins in nutrient-rich broth.

*Mtb* Gene	*Mav* Gene	*Mtb* LFC	*Mav* LFC	*Mtb* Annotation	*Mav* Annotation
*rv0049*	*DFS55_00355*	−2.99	−1.09	hypothetical protein	hypothetical protein
*rv0819*	*DFS55_21365*	1.33	1.23	mycothiol acetyltransferase	mycothiol synthase
*rv0820*	*DFS55_21345*	−1.42	−1.69	phosphate ABC transporter ATP-binding protein PhoT	phosphate ABC transporter ATP-binding protein
*rv0929*	*DFS55_20215*	−1.23	−1.87	phosphate ABC transporter permease PstC	phosphate ABC transporter permease PstC
*rv0930*	*DFS55_20210*	−1.16	−1.00	phosphate ABC transporter permease PstA	phosphate ABC transporter permease PstA
*rv1836c*	*DFS55_12730*	−1.45	−1.44	hypothetical protein	hypothetical protein
*rv2179c*	*DFS55_14810*	−2.72	−1.09	3′-5′ exoribonuclease A	hypothetical protein
*rv2224c*	*DFS55_15065*	−1.80	−2.17	carboxylesterase A	alpha/beta hydrolase
*rv3005c*	*DFS55_07355*	−1.25	−1.06	hypothetical protein	hypothetical protein

**Table 2 microorganisms-11-02269-t002:** *Mtb* genes implicated specifically in starvation-induced tolerance to rifampin.

Gene	7H9 6d LFC	PBS 7d LFC	PBS 14d LFC	Annotation
*rv0458*	−0.11	2.30	1.05	aldehyde dehydrogenase
*rv0819*	1.33	−1.44	−1.67	mycothiolacetyl transferase
*rv0989c*	0.03	2.77	1.01	polyprenyl-diphosphate synthase GrcC
*rv0998*	−0.22	−1.24	−1.25	acetyltransferase Pat
*rv1183*	0.1	0.52	1.12	transmembrane transport protein MmpL10
*rv1908c*	0.85	−0.88	−1.15	catalase-peroxidase
*rv2051c*	−0.19	−1.33	−1.23	polyprenol-monophosphomannose synthase
*rv2199c*	−0.5	2.53	2.42	cytochrome c oxidase polypeptide 4
*rv2374c*	−0.2	−1.47	−1.68	heat-inducible transcription repressor HrcA
*rv2392*	0.95	−1.29	−1.17	phosphoadenosine phosphosulfate reductase
*rv2633c*	0.41	1.76	1.02	hypothetical protein
*rv2709*	1.16	−1.26	−1.16	transmembrane protein
*rv2733c*	0.01	−1.55	−1.19	(Dimethylallyl)adenosine tRNA methylthiotransferase
*rv3680*	−0.26	−1.12	−1.40	anion transporter ATPase
*rv3923c*	−0.17	−1.1	−1.36	ribonuclease P protein component

**Table 3 microorganisms-11-02269-t003:** *Mtb* mutants with differential susceptibility to rifampin in both nutrient-rich (7H9) and nutrient-starved (PBS) conditions.

Gene	7H9 6d LFC	PBS 7d LFC	PBS 14d LFC	Annotation
*rv0049*	−2.99	−1.45	−1.81	hypothetical protein
*rv0199*	1.14	4.50	2.53	membrane protein
*rv0200*	1.61	4.69	2.56	transmembrane protein
*rv0655*	1.32	5.03	3.01	ABC transporter ATP-binding protein
*rv0819*	1.33	−1.44	−1.67	mycothiol acetyltransferase
*rv0994*	−3.25	−1.27	−1.01	molybdopterin molybdenumtransferase 1
*rv2179c*	−2.72	−1.32	−1.30	3′-5′ exoribonuclease
*rv2690c*	5.56	3.26	1.63	integral membrane protein
*rv2709*	1.16	−1.26	−1.16	transmembrane protein
*rv3723*	1.18	4.94	3.04	transmembrane protein

## Data Availability

The raw sequencing data (*.fastq) from this project can be at NCBI under BioProject PRJNA946182. Jupyter notebooks and associated scripts to reproduce the data analysis from the raw data are provided online at https://doi.org/10.5281/zenodo.7792160 (accessed 8 September 2023).

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
