# Peer review of "Functional Whole Genome Screen of Nutrient-Starved Mycobacterium tuberculosis Identifies Genes Involved in Rifampin Tolerance"

_microorganisms, 2023, doi:10.3390/microorganisms11092269_

Round 1

Reviewer 1 Report

This study presents a highly promising study with critical implications for tuberculosis treatment. The thorough exploration of Mycobacterium tuberculosis's adaptation mechanisms to antibiotic stress and nutrient conditions is impressive. Identifying specific genes linked to antibiotic tolerance under varying environments demonstrates a comprehensive approach. The potential therapeutic targets highlighted in this research have the potential to improve antitubercular regimens, addressing the pressing need for more efficient treatments. The writing is professional and easy to understand, and the references are reasonable. Overall, the study's strategic design and significant findings make it a valuable contribution to the field.

Some minor comments and suggestions:

1. What criteria were used to select the concentrations of isoniazid and rifampin for the screening process? i.e., the concentrations of rifampin in lines 298 and 299.

2. Have the authors ever tried to examine the effects of isoniazid on the four identified mutants (ercc3, moeA1, Rv0049, and Rv2179c)?

3. Have the authors performed the screening with rifampin and isoniazid combination? This might be an interesting assay.

4. How will the identifications of these four mutants contribute to the development of new therapies needs more discussion.

5. Since most of the contents and characterizations in the manuscript were rifampin-dependent. I would suggest replacing “antibiotic” with “rifampin” in the title.

Author Response

Response to Reviewer 1 Comments

1. What criteria were used to select the concentrations of isoniazid and rifampin for the screening process? i.e., the concentrations of rifampin in lines 298 and 299.

We appreciate the Reviewer’s comment. We have added explanations of how the concentrations were selected and which concentrations were used for analysis to the Methods sections (lines 145-149 and 166-173).

2. Have the authors ever tried to examine the effects of isoniazid on the four identified mutants (ercc3, moeA1, Rv0049, and Rv2179c)?

We thank the Reviewer for this suggestion. At this time, we have not tested the susceptibility of these individual mutants to isoniazid as the Tn mutant screen in isoniazid did not reveal altered phenotypes for mutants in these genes.

3. Have the authors performed the screening with rifampin and isoniazid combination? This might be an interesting assay.

We appreciate the Reviewer’s comment. Although we have not performed such studies, Block et al previously published a screen using the combination of rifampin and isoniazid (PMID: 36598240), which we cite and compare to our results in the Discussion.

4. How will the identifications of these four mutants contribute to the development of new therapies needs more discussion.

We thank the Reviewer for the comment. We humbly acknowledge much work needs to be done before small-molecule inhibitors of these gene products could be developed as novel therapies. However, we have elaborated on the potential importance of these novel targets in the revised Discussion (Line 362 -366).

5. Since most of the contents and characterizations in the manuscript were rifampin-dependent. I would suggest replacing “antibiotic” with “rifampin” in the title.

We agree with the Reviewers and have changed the title accordingly.

Reviewer 2 Report

Dear Authors,

The manuscript entitled "Functional whole genome screen of nutrient-starved Mycobacterium tuberculosis identifies genes involved in antibiotic tolerance" is an interesting study about "the identification of candidate Mtb genes involved in tolerance to the two key first-line antibiotics, rifampin, and isoniazid, under nutrient-rich and nutrient-starved conditions".  

I only suggest adding an explanation to the introduction about other alternative therapies such as phage therapy (Line 41).

Author Response

Response to Reviewer 2 Comments
The manuscript entitled "Functional whole genome screen of nutrient-starved Mycobacterium tuberculosis identifies genes involved in antibiotic tolerance" is an interesting study about "the identification of candidate Mtb genes involved in tolerance to the two key first-line antibiotics, rifampin, and isoniazid, under nutrient-rich and nutrient-starved conditions".  
I only suggest adding an explanation to the introduction about other alternative therapies such as phage therapy (Line 41).

We thank the Reviewer for the favorable comments. We appreciate the Reviewer’s suggestion and have referred to alternative therapies such as phage therapy in the revised Introduction (lines 47-48).